# Polygalaxanthone III, an Active Ingredient in *Polygala japonica Houtt.*, Repaired *Malassezia*-Stimulated Skin Injury via STAT3 Phosphorylated Activation

**DOI:** 10.3390/molecules27217520

**Published:** 2022-11-03

**Authors:** Xiaobin Yang, Bei Xiong, Zhuolei Yuan, Hui Liao, Xiaowei Liu, Yinan Wu, Shu Zhang, Qi Xiang

**Affiliations:** 1Institute of Biomedicine and Guangdong Provincial Key Laboratory of Bioengineering Medicine, Jinan University, Guangzhou 510632, China; 2Guangdong Provincial Key Laboratory of Advanced Drug Delivery Systems, Center for Drug Research and Development of Guangdong Pharmaceutical University, Guangzhou 510006, China; 3Biopharmaceutical R&D Center, Jinan University, Guangzhou 510632, China; 4National Engineering Research Center of Genetic Medicine, Guangzhou 510632, China

**Keywords:** *Malassezia*, skin injury, Polygalaxanthone III, p-STAT3

## Abstract

*Malassezia* is a genus of commensal and lipid-dependent yeasts in human skin which also have a pathogenic lifestyle associated with several common skin disorders such as atopic dermatitis and eczema. Symptoms include red, itchy, and inflamed skin. We studied the growth characteristics and biochemical analyses of *M. furfur* which showed that the protein contents were greater in extracts taken at 24 h. These were then used to infect C57BL/6 mice, resulting in skin rupture. Polygalaxanthone III (POL), a more effective anti-inflammatory ingredient in *Polygala japonica Houtt.*, was applied externally to the ulceration and successfully healed the wounds quickly. POL could not inhibit *Malassezia* activity as tested by the inhibition zone test, but affected the formation of lipid droplets in HaCaT cells. The wound-healing molecular mechanisms may be involved in the STAT3 pathway according to the Western blot results of skin tissues. *Malassezia*’s role in skin health is far from certain, and there is no clear solution, so understanding the development of *Malassezia*-associated skin diseases in general and seeking solutions are very important.

## 1. Introduction

The *Malassezia* genus comprises single-celled fungi belonging to the yeast group, some of which are the main genus of fungi on human skin, and can be further divided into *Malassezia furfur* (*M. furfur*), *Malassezia pachydermatis* (*M. pachydermatis*), *Malassezia globose* (*M. globosa*), *Malassezia sympodialis* (*M. sympodialis*), etc. [1]. *Malassezia* lacks fatty acid synthase and thus is reliant on synthesizing lipases to decompose lipids to obtain fatty acids [1,2] for its growth. *Malassezia* is mostly found in sebum-rich parts of the skin such as the scalp, face, and back. *Malassezia* yeasts are conditionally pathogenic bacteria [3,4]; when the skin humidity, pH, and temperature change, these yeasts can cause skin diseases such as *Malassezia* folliculitis and seborrheic dermatitis [5]. Once infected by *Malassezia*, skin tissues are generally affected by acute inflammation, resulting in persistent and recurrent damage; thus, it has a serious impact on the quality of life of patients. However, only Amphotericin B is recommended to treat skin diseases caused by *Malassezia* in clinics [3,6]. Long-term use of antifungal drugs is associated with a certain degree of tolerance, low clinical safety, and poor therapeutic efficacy, and has become a pressing issue in current research [7,8,9]. Thus, it is very important to understand the development of *Malassezia*-associated skin diseases in general and to find solutions.

Over the last two decades, together with exploring the human microbiome, *Malassezia* has moved from virtually unknown niche research into a key global focus area [10,11]. Many studies have suggested that the pathogenesis of *Malassezia* is related to alterations in the skin microenvironment or defense function [12]. Overpopulation of *Malassezia* synthesizes too much lipase, which breaks down lipids into fatty acids causing skin irritation and inflammation [13] and inflamed skin exacerbates *Malassezia* pathogenicity [14]. The traditional Chinese medicine, *Polygala japonica Houtt.* (PJ), is a member of Polygalaceae family. In China, PJ has been used externally to treat diseases such as furuncle, bruises, injuries, and snake and insect bites [15]. Furuncle is a kind of skin inflammatory disease caused by fungal or bacterial infection of hair follicles or sebaceous glands and is directly related to *Malassezia*. In our previous study, an anti-inflammatory active ingredient of Polygalaxanthone III (POL) from *Polygala japonica Houtt.* was screened out, which downregulates inflammation in the lipopolysaccharide-stimulated RAW264.7 macrophages [16]. Thus, we are trying to use it to treat skin damage caused by *Malassezia*.

As a viable alternative to synthetic drugs, traditional Chinese medicine (TCM) is characterized by few side effects, wide availability, and low drug resistance. The treatment of TCM in regulating skin flora and reestablishing micro-ecological balance has unique advantages. In this paper, we employed animal models to confirm the effectiveness of POL in *Malassezia*-mediated wound-healing enhancement and provided a solution to treat *Malassezia*-associated skin diseases. Our research reflects the irreplaceable clinical, market, and scientific value of TCM [17,18].

## 2. Results

### 2.1. Growth Curve and Characteristics of the Extracts of M. furfur

The growth curve of *M. furfur* is shown in Figure 1A. It enters a rapid growth phase from 12 h to 24 h of culture, and then a stable growth phase is observed. The results of different phases of culture supernatant and bacterial pellet growth (12 h, 24 h, 36 h, 48 h, and 60 h) tested by SDS-PAGE are shown in Figure 1B. There are five protein bands that can be observed by the naked eye in the supernatant: the protein band around 100 KD gradually increased significantly and changed with time, the band around 50 KD gradually weakened, and the other three bands gradually appeared near 38 KD and 28 KD. There are more than 10 protein bands in the cell protein, around 100 KD, 70 KD, 50 KD and 38 KD.

The SDS-PAGE protein gel was tested and stained, and the protein bands at 50 KD and 70 KD were the most obvious. Mass spectrometry analysis reveals the 70 KD protein A is unnamed proteins A0A2N1JC25 (Protein ID: A0A2N1JC25, C25), and the 50 KD protein B is secreted lipase 2 (Protein ID: E1CJK0, LIP2).

### 2.2. Malassezia furfur-Infected C57BL/6 Mouse Skin Resulted in Skin Rupture

After infection for approximately 1 to 2 days, the infected part of the skin began to redden and swell, causing rupture. The wound took about 14 days to heal naturally (Figure 2). From the picture of HE, the pathological features of infected C57BL/6 mouse skin were evident, that is, with blurred junction lines between epidermis and dermis, obvious granulation tissue, disorganized, deposited, and increased collagen fibrils, still a large number of inflammatory cell infiltrates, and blurred boundaries of sebaceous glands at the injury site. The results of Masson staining showed that the hair follicles were reduced and the fat layer was thickened. The results of oil red O staining showed that the lipid drops were increased and the distribution was not uniform, mainly concentrated in both sides of the injured site.

### 2.3. Polygalaxanthone III Festinating Repaired the Malassezia furfur-Infected Skin Injury

The daily changes in the skin wounds of the different groups were observed visually (Figure 2). Compared to the model group, the four aspects of wound size, degree of injury, crusting, and healing time were significantly improved after treatment with drugs. The wounds were significantly controlled after POL external administration compared to the blank vehicle group, with not only a smaller area of crusting but also a shorter time required for healing (vs. blank vehicle group). When comparing the high- and low-dose POL groups with the PJ group, the application of POL showed greater improvement in wound size, degree of damage, crusting, and healing time, with the wounds almost smoothly healed in 6 days without scarring.

In summary, the changes in the wounds induced by *M. furfur* infection led to a preliminary conclusion that POL had a reparative effect on the injury, taking less time for the skin to heal and significantly improving the crusting.

### 2.4. Histologic Examination and Masson Staining

Figure 3A shows the HE staining results for different groups of whole skin samples containing the entire wound site on day 6 and day 12. In the skin tissue on day 6, some granulation tissue was visible in the HE staining of the trauma samples from the different treatment groups compared to the normal group and showed varying degrees of thickening of the epidermal layer.

In the PJ group and the low- and high-dose POL group, the epidermis was thicker than in the blank vehicle group, but the boundary between the epidermis and the dermis is obvious, and the collagen fibers appeared to be more regular than in the model group, especially in the low-dose POL group. A small infiltration of inflammatory cells was also observed in both the blank vehicle group and the high-dose POL group, with individual collagen fibers arranged in a swirling, disordered fashion.

In the samples on day 12, both HE staining and Masson staining results showed that each group was better than on day 6, with a clear epidermal–dermal junction line, fewer inflammatory cells, and more regular collagen fibers. The Masson staining provided a clearer and more visual picture of the arrangement of collagen fibers in the specimens (Figure 3B). Overall, some granulation tissue could still be observed in the model group, and the epidermal thickness was observed (with the naked eye) to increase compared to the normal group. A small amount of inflammatory cell infiltration was still visible in the simple skin injury group, the model group, and the blank vehicle group. Compared with the model group, the epidermal thickness of the PJ and POL groups was significantly reduced, and the collagen distribution was regular and no inflammatory cell infiltration was seen, similarly to the normal group, indicating that the wounds had been completely repaired and there was no obvious inflammatory reaction.

At the end of the experiment, all the POL groups had no obvious granulation tissue, epidermis and dermis boundaries were clear, there was no obvious thickening, collagen fibers were neatly arranged and orderly, and sebaceous gland morphology was arranged in normal, clear boundaries. This shows that POL application repaired skin damage better.

### 2.5. Identification and Microscopic Examination of Malassezia furfur Colonies in Wound Skin

After the cotton swab extracts were cultivated for 48 h, some single colonies were grown in the mDixon plates (Figure 4A). The single colonies in the plates can be divided into two categories according to their morphological size: one category of single colonies is small, with a diameter of about 1 mm, while the other category of single colonies is relatively large, with a diameter of about 2 mm. The results of colony PCR are shown in Figure 4B, where 1–9 were colonies from normal skin, 10–20 from the skin damage group, 21–23 from the molding group, 24–33 from the blank cream group, 34–39 from the guaifenesin cream group, 40–49 from the POL low-dose cream group, 50–55 from the POL high-dose cream group, and 57 was used as a negative control (sterilized water was used as a PCR template). The single colonies selected from all of the low-dose and high-dose POL groups were negative, but the model, blank vehicle, and PJ groups were positive. Further microscopic examination of the positive single colonies was carried out. As can be seen in Figure 4C, the microscopic purple color represents elevated, button-shaped colonies, presenting a morphology.

There was an obvious inhibition zone in the outer circle of the positive drug G418 group, indicating that G418 can inhibit the growth and that different concentrations of the POL treatment group had no obvious inhibition zone (Figure 5C). It can be concluded that POL has no obvious inhibitory effect on the growth.

### 2.6. POL Reduced the Amount of Lipid Droplets Secreted by M. furfur-Stimulated HaCaT Cells

The results of cells pre-treated with POL for 6 h are shown in Figure 5A. There was no significant difference in the content of lipid droplets between the POL group and the model group (*p* > 0.05), and there is no significant difference in the NO content between them (*p* > 0.05). The results of cells irrupted following POL treatment, which showed that the intracellular lipid droplet content was significantly reduced when POL was administered at concentrations greater than 0.5 mM vs. the model group (*p* < 0.001). Treatment with POL was effective in reducing the amount of lipid droplets secreted by *M. furfur*-stimulated HaCaT cells.

### 2.7. Western Blot Analysis of Mice Skin Tissues

The changes in the JAK-STAT signaling pathway-related proteins in the traumatic skin tissue of each group were detected by Western blot, and the results are shown in Figure 6. From the graphs, the expressions of JAK1 are different in the groups, but the phosphorylation of JAK1 (p-JAK1) cannot be detected in all experimental groups. p-JAK2 in model group cannot be detected, either. However, after electric frosting using a roller, the ratio of p-JAK2/JAK2 significantly decreased (*p* < 0.01 vs. normal group), and the ratio of p-JAK2/JAK2 was significantly upregulated after treatment with POL. The expressions of STAT1, p-STAT1, and p-JAK2/JAK2 also had no significant changes in other groups except the normal group.

Among them, the changes in p-STAT3 and STAT3 are notable. The expressions of p-STAT3 and STAT3, and the ratio of p-STAT3/STAT3 were significantly decreased in the model group (*p* < 0.05, vs. normal group), but after treatment with POL groups, all of them were upregulated (*p* < 0.05, vs. model group).

## 3. Discussion

### 3.1. The Excess of M. furfur Damaged Skin Barrier Function

As an important component of the skin flora, *Malassezia* parasitizes mainly in the vicinity of sebaceous glands, where lipids are essential for their development. Studies of the *Malassezia* genome have revealed that all species of the genus lack the gene coding for fatty acid synthase [19], which therefore means that these yeasts require lipids to grow. During the growth process, *Malassezia* will secrete large amounts of lipase to disintegrate lipids in an ambient environment into fatty acids in order to provide the necessary nutrients for its survival. We cultured *M. furfur* in vitro and found that it entered the logarithmic growth phase after 12 h and the stationary phase after 24 h; the proteins separated by SDS-page showed that the kind and the expression of proteins in *M.furfur* varied in different growth stages. Combining the OD_600_ values and the amounts of protein, we optimized the dose of *M. furfur* for further experimentation.

The two most expressed proteins we identified in the *M. furfur* cells are C25 and LIP2. LIP2 is thought to be associated with lipid metabolism. The protein expression in the supernatant of *M. furfur* was relatively low, with the most expressed protein at around 50 KD at 12 h, the same position as the identified LIP2, and the most expressed protein at around 100 KD at 24 h, just twice as high as LIP2, possibly a polymer of LIP2. The ability of *M. furfur* to metabolize lipids and integrate fatty acids into its cell wall is required for growth and survival in the host environment, which significantly increases the pathogenicity of *M. furfur* [20]. Future work is needed to explore the effects of LIP2 on the skin barrier and subcutaneous lipid-associated cells in mice.

According to the literature, *Malassezia* colonize deep inside the skin. To facilitate *Malassezia* entry into the skin and avoid traumatic lesions, we first made a minimally invasive skin lesion by lightly rubbing the skin on the back of C57BL/6 mice using an electric scrub roller, then followed by the uniform topical application of an olive oil suspension containing *M. furfur*. The scheme thus achieved the model of *Malassezia*-infected C57BL/6 mice. In addition to electric abrasive rollers, sandpaper is often used to disrupt the skin barrier prior to experimental infection of mice with S. aureus or Candida albicans [21,22]. Compared to sandpaper, the electric abrasive roller provides better control of the force and extent of trauma. Thus, we applied both electric frosting roller and *M. furfur* suspension to create the skin injury model.

### 3.2. POL Attenuated Lipid Droplet Formation in M. furfur-Stimulated HaCaT Cells

*Malassezia* parasitic in the human body can interact with the host. Studies have shown that *Malassezia* mainly interacts with keratinocytes, dendritic cells residing in tissues, macrophages, and myeloid cells, and myeloid cells are recruited into the skin under inflammatory signals [23,24]. At the same time, extracellular vesicles produced by *Malassezia* can activate human primary keratinocytes to enhance the expression of cells to defense against *M. sympodialis* [25]. Therefore, based on the dependence of *Malassezia* on lipids and interaction with keratinocytes, we investigated the effects of POL on HaCaT cells, especially on lipid droplet formation in HaCaT cells. It was found that the amount of lipid droplets secreted by HaCaT cells was dose-dependent on *Malassezia* supernatant. When using treatment-stimulated HaCaT cells with POL, the lipid droplet contents significantly reduced in cells (*p* < 0.001), but the extracellular NO content was unchanged, which was somewhat surprising. In vitro inhibition assays showed that POL did not inhibit the growth of *Malassezia*. Therefore, we hypothesized that the reduction in lipid droplets might affect the survival environment of *Malassezia*, thus reducing the reproduce of *Malassezia* indirectly. In *M. furfur*-infected C57BL/6 mice, we identified *M. furfur* colonies in wound skin, but not in POL-treated mice, which would be evidence of our hypothesis.

That is, we think POL does not have a direct effect on the growth of *M. furfur*, but has an inhibitory effect on the lipid droplet production stimulated by it, thus restricting multiplication of *M. furfur* and reducing its potential harm.

### 3.3. POL Repaired Malassezia furfur-Stimulated Skin Injury via STAT3 Phosphorylated Activation

In certain pathological conditions, the inflammatory process is widely recognized as a localized protective response of the body in cases such as pathogen attacks or damages. The JAK/STAT is the primary signaling pathway regulated by cytokines and is crucial for initiating the innate immunity, orchestrating the adaptive immune mechanisms, and finally constraining the inflammatory and immune responses [26,27,28].

Our previous work had already suggested that POL would downregulate inflammation in lipopolysaccharide-stimulated RAW264.7 macrophages via the JAK-STAT pathway. The JAK/STAT signaling pathway is evolutionarily conserved. It is composed of ligand-receptor complexes, JAKs, and STATs [29]. In order to verify the molecular mechanism of POL repairing the *M. furfur*-stimulated skin injury, the expression of JAK-STAT pathway-associated proteins was tested. Firstly, we detected the expression of JAKs with protein switches that activate other proteins by adding a phosphate group to them via phosphorylation. Unfortunately, p-JAK1 cannot be detected in every group. Then, STATs were detected. Among them, the expression of p-STAT3 and STAT3 are upregulated after treatment with the POL and PJ groups compared with the model group (*p* < 0.05). According to the report of Jean-Laurent Casanova et al., the mutation of SATA3 would result in susceptibility to fungal infections, which means STAT3 would be activated independent of JAK [30].

The high degree of allelic heterogeneity at the human STAT1 and STAT3 loci has revealed highly diverse immunological and clinical phenotypes [31,32,33]. Furthermore, skin barrier function is regulated via competition between the aryl hydrocarbon receptor (AHR) axis (up-regulation of barrier) and the IL-13/IL-4-JAK-STAT6/STAT3 axis (down-regulation of barrier) [34]. This latter axis also induces oxidative stress, which exacerbates inflammation. The AHR upregulates IL-1 receptor type 1 (IL-1R1) expression in Th17 cells. The AHR also inhibits signal transducer and activator of transcription (STAT)1 and STAT5, which negatively regulates the Th17 program, and together with STAT3, induced Aiolos expression that resulted in IL-2 silencing. In Tr1 cells, the AHR interacts with musculoaponeurotic fibrosarcoma (c-Maf) to induce the expression of IL-10 and IL-21 with STAT3 to drive CD39 expression and its own expression, which acts as a positive feedback loop. This process involving AHR results in complex schemas [35]. Sparber et al. discussed the immune response to *Malassezia* and the way in which the implicated cells and cytokine pathways prevent uncontrolled fungal growth to maintain commensalism in mammalian skin [36]. There are deep and challenging questions in areas that are impossible for us to illustrate clearly with limited experimental data. We will continue to do so in the years to come.

On the other hand, the expression of p-STAT1/STAT1 was not significantly different (Figure 6A) [37,38]. The results are inconsistent with our cytological experiment using lipopolysaccharide-stimulated RAW264.7 macrophages. Firstly, we think that the lipopolysaccharide is produced from the bacteria, but *Malassezia* is a fungus, so the results do not correspond with each other. Secondly, in vitro animal tissue experiments are considerably complicated compared to cell tissue experiments and different interpretations are possible.

To sum up, we theorized that POL would accelerate the repair of the *M. furfur*-stimulated skin injury via STAT3 phosphorylated activation independent of JAK and associate with the immune response.

## 4. Materials and Methods

### 4.1. Chemicals and Reagents

POL was purchased from Tianzhi Biological (Wuhan, China). Furthermore, 0.25% trypsin-ethylenediaminetetraacetic acid (EDTA), RIPA lysis buffer (Beyotime Biotechnology, Shanghai, China) formaldehyde and pentanediol were purchased from Guangzhou Chemical Reagent (Guangzhou, China). Then, 2693 modified Dixon medium (mDixon, 36 g/L Malt Extract, 20 g/L Desiccated Ox bile, 6 g/L Pepton, 2 g/L Glycerol, 2 g/L Oleic Acid, 10 mL/L Tween 40, pH 6.0 ± 0.1) was purchased from Haibo Biological (Qingdao, China). Gram staining solution was purchased from Huankai Microbe (Guangzhou, China). Depilatory cream was purchased from VEET (Jingzhou, China) and chromogenic medium of *Malassezia* spp. was purchased from Chromagar (Paris, France).

### 4.2. Strains

*Malassezia furfur* (ATCC^®^ 14521TM, *M. furfur*) was purchased from Guangdong Institute of Microbiology (Guangzhou, China) and grown in a 37 °C incubator with mDixon.

### 4.3. Cells

A homogeneous HaCaT cell line (RRID: CVCL_0038) was purchased from the Chinese Academy of Sciences (Shanghai, China). It was derived from a common ancestor of the HaCaT cell line and verified by Guangzhou Cellcook Biotech Co., Ltd. (Guangzhou, China). The cells were cultured in DMEM supplemented with 10% FBS, 100 U·mL^−1^ penicillin and 100 μg·mL^−1^ streptomycin in a humidified incubator at 37 °C and 5% CO_2_.

### 4.4. Animals

SPF grade C57BL/6 mice (approximately 8 to 12 weeks, 20 ± 2 g, half male and half female) were purchased from Guangdong Experimental Animal Center (Guangzhou, China) (Certificate No. 44007200086896) and all mice were housed in a 25 °C and approximately 40% to 70% humidity laboratory animal room. This room was cycled in 12 h of light and 12 h of darkness every day, and the mice drank and ate freely. The mice experiments were approved by the animal ethics committee of Jinan University and strictly abide by the principles of welfare and protection of experimental animals (approval number: 2019228).

### 4.5. Growth Curve and Biochemical Analyses of Malassezia furfur

The growth curve of *M. furfur* was drawn based on the changes in absorbance during the growth of *M. furfur*. Furthermore, 150 mL mDixon was distributed equally into 3 flasks (250 mL) and sterilized by autoclaving at 121 °C for 15 min. A 1 mL *M. furfur* bacteria liquid (10^6^ cells/mL) was transferred into the above flasks and shaken for 60 h at 37 °C, 200 rpm. During the cultivation process, 1 mL *M. furfur* solution was taken from each flask, first at 12 h and every 12 h thereafter. The solution was divided into 2 parts. One part was used to measure the absorbance of λ = 600 nm (OD_600_) to draw the growth curve. The other part was centrifuged at 8000 rpm for 8 min; then, the supernatant was separated and precipitated for SDS-PAGE. The cutting of gel bands and extracting of proteins was performed by using a micro protein PAGE recovery kit (Beijing Solarbio Science & Technology Co., Ltd., Beijing, China); then, the proteins were sent to the Beijing Genomics Institute (BGI, Beijing, China) for LC-MS/MS analysis.

### 4.6. Methods for LC-MS/MS Analysis

We commissioned Beijing Biotech-Pack Co. Ltd. (Beijing, China) to conduct mass spectrometry analysis. Proteins A and B recovered from the SDS-page gel were digested by trypsin and an Ultimate 3000 system (Thermo Fisher Scientific, Waltham, MA, USA) interfacing with a Q Exactive™ Hybrid Quadrupole-Orbitrap™ Mass Spectrometer system (Thermo Fisher Scientific, Waltham, MA, USA) was used for the LC-MS/MS analysis. The peptides were reconstituted with 10 μL of 0.1% formic acid (0.1% FA, 2% CAN), and 5 μL of the solution was loaded and separated on a 150 μm × 15 cm in-house-made column packed with a reversed-phase ReproSil-Pur C18-AQ resin (1.9 μm, 100 Å, Dr. Maisch GmbH, Ammerbuch-Entringen, Germany). The mobile phase for chromatographic separation consisted of 0.1% formic acid in 2% acetonitrile (A) and 0.1% formic acid in 80% acetonitrile (B). The flow rate was set to 600 nL/min. The gradient was set as follows: 0–5 min, 6–9% B; 5–20 min, 9–14% B; 20–50 min, 14–30% B; 50–58 min, 30–40% B; 58–60 min, 40–95% B. The MS spectra were acquired from 350 to 1800 *m*/*z* with a resolution of 70,000 FWHM, and MS/MS scan resolution of 75,000. The top 20 most intense peptide ions from the preview scan in the Orbitrap were selected. The raw MS files were analyzed and searched against a target protein database based on the species of the samples using MaxQuant (1.6.2.10).

### 4.7. Animal Model and Experimental Protocol

A total of 28 SPF grade C57BL/6 mice were anesthetized by i.p. 2% Pentobarbital, then the back hair was shaved, followed with the application of depilatory cream. All mice were randomly divided into 7 groups (2 male and 2 female mice in each group): (A) normal group, (B) electric frosting roller group, (C) model group (electric frosting roller and *M. furfur* suspension), (D) blank vehicle group, (E) Polygala japonica Houtt group (1.25% *Polygala japonica Houtt.* extract, W/W), (F) low-dose POL group (POL L, 1% POL in vehicle, *w*/*w*), (G) high-dose POL group (POL H, 2% POL in vehicle, *w*/*w*).

After hair removal for 24 h, two areas with a size of 1 × 1 cm were marked on each side of the back, for a total of 8 wounds in each group. The modeling area was disinfected and rubbed with an electric frosting roller (Ruiward Life Technology Co., Ltd., Shenzhen, China) until small spots of blood appeared on the skin. The *M. furfur* that was cultivated for 24 h was centrifuged and resuspended in olive oil, diluting the suspension until the absorbance value of OD_600_ was 0.4 (measured value after 10 times dilution), then groups C~G were treated with 50 μL *M. furfur* suspension. When the wound surface was dry, D~G groups were treated once every day with 0.04 g drug products. We observed the condition of all mice, weighed them every day, and took pictures to record the degree of wound healing.

On the 7th and 14th day after the mice were treated with drugs, 2 mice (1 male and 1 female) were randomly selected, and their wounds were swabbed for *M. furfur* identification. After euthanizing the mice, the skin tissues of the treated area were collected and divided into two halves, one frozen at −80 °C for later assays, another fixed in 10% formal and embedded in paraffin for HE and Masson staining analysis.

### 4.8. Identification of Malassezia furfur

The cotton swab extracts were first cultured with mDixon, then the positive colonies were selected to culture in chromogenic medium of *Malassezia* spp. in which fold-packed colonies would turn pale pink if they were *M. furfur*.

The selected positive colonies on the mDixon plate, after being resuspended with ddwater, were Gram stained (Guangdong Huankai Microbial Sci&Tech. Co., Ltd., Guangzhou, China) and observed under a Nikon ECLIPSE NI-E (Nikon Co, Tokyo, Japan).

Colony PCR was performed for further confirmation with a rapid PCR machine (Langji Scientific Instruments Co. Ltd., Hangzhou, China) (F:5′-*CTCGCGTACAACGTCTCTGG*-3′, R: 5′-*CGCTGCGTTCTTCATCGA*-3′).

### 4.9. Western Blotting

The proteins were extracted from about 20 mg of the mouse wound skin tissue, and GAPDH was used as the internal reference protein to detect the expression levels of JAK2, p-JAK2, STAT1, p-STAT1, STAT3, and p-STAT3.

We took about 20 mg of frozen mouse wound skin tissue collected on the 7th and 14th days into a 1.5 mL centrifuge tube; then, 200 μL RIPA lysis buffer containing protease inhibitors and stainless steel balls was added to it separately, and it was grinded (60 Hz/min, 3 min) immediately with an automatic fast-speed tissue grinder (Shanghai Jingxin, Shanghai, China). Tissue lysate was centrifuged at 14,000 rpm for 30 min at 4 °C (Sigma 3k-15, Sigma, Osterode am Harz, Germany), then the supernatant was collected. The supernatant protein concentration was determined by BCA protein assay kit (Thermo, Waltham, MA, USA), 50 mg protein sample was loaded onto a 10% SDS-PAGE for electrophoresis, and then transferred to PVDF membranes (W016-1-1, Nanjing Jiancheng Bioengineering Research Institute, Nanjing, China). Next, skimmed milk powder was sealed for 2 h, and Western blot analysis of protein expression was performed. The immunoreactive bands in the blots were quantified using Image J software (NIH, Bethesda, MD, USA).

The following primary antibodies were used: anti-JAK2 (ABclonal Technology, Wuhan, China; 1:500), anti-p-JAK2-Y1007/1008 (ABclonal Technology, Wuhan, China; 1:500), anti-STAT1 (Proteintech Group, Beijing, China; 1:2000), anti-p-STAT1-Y701 (ABclonal Technology, Wuhan, China; 1:1000), anti-STAT3 (Proteintech Group, Beijing, China; 1:2000), anti-p-STAT3-S727 (ABclonal Technology, Wuhan, China; 1:1000), and anti-GAPDH (Bioworld Technology, Inc., Bloomington, MN, USA; 1:1000). The primary antibodies were detected using the horseradish peroxidase-conjugated secondary antibody (Ford Biotechnology Co., Ltd., Hangzhou, China; 1:5000).

### 4.10. Lipid Droplets Formed in HaCaT Cells

The HaCaT cells in the logarithmic growth phase were plated in a 12-well plate (2 × 10^6^ cells/mL). To begin with, we detected the effects of *M. furfur* solutions irritating HaCaT cells and screened out the optimum concentration of *M. furfur* solutions. The experimental groups were as follows: the normal HaCaT cells group (control group), *M. furfur*-irritated group (model group), dexamethasone group (DXMS, positive control group), and different concentrations of POL groups. To test the properties of POL, two modes of administration were employed. In brief, for the first mode, after the cells’ attachment, the medium was removed, cells were pretreated with POL for 6 h, and then cells were irritated with *M. furfur* for 24 h. The other mode involved cells being irritated with *M. furfur* solutions for 6 h and then treated with POL for 24 h. After the drug treatment, the cell morphology and lipid droplets in cells were observed and photographed under an inverted microscope (Guangzhou Mingmei Photoelectric Technology Co., Ltd. Guangzhou, China). The cell culture supernatant was collected to detect the NO content, and the lipid droplets were stained with oil red O.

### 4.11. Statistical Analysis

All experimental operations were repeated at least 3 times, and GraphPad Prism 8.0 software was used for data processing. The experimental data were expressed as “mean ± standard deviation”, and the difference between the data of each group was compared by *t* test or one-way analysis of variance. * *p* < 0.05 and ** *p* < 0.01 indicate that there are statistical differences and significant statistical differences, respectively.

## 5. Conclusions

Eczema, seborrheic dermatitis (SD), and so on are the result of the hydrolysis of free fatty acids and activation of the immune system through pattern recognition receptors, inflammasomes, interleukin-1β, and nuclear factor-κB [39]. The interaction of Malassezia, keratin-forming cells, and the immune response against the altered lipid composition in the skin plays a crucial role in the pathogenesis of Malassezia dermatoses. Although the sequence of pathophysiological events relating to them is unclear, most sources consider the three main prerequisites to be Malassezia colonization, sebaceous gland lipid secretion, and underlying immune system susceptibility [40]. The pathogenesis can be divided into five distinct phases [41]: sebaceous glands secrete lipids onto the skin surface; Malassezia colonizes areas covered with lipids; Malassezia secretes lipase, which produces free fatty acids and lipid peroxides that activate the inflammatory response; the immune system produces cytokines such as IL-1α, IL-1β, IL-2, IL-4, IL-6, IL-8, IL-10 IL-12, and TNF-α, which stimulate keratinocyte proliferation and differentiation; and disruption of the skin barrier, leading to clinically evident erythema, pruritus, and scaling.

This study successfully established a mouse skin model of infection and found that POL can accelerate wound healing via STAT3 phosphorylated activation, which plays an important role in fungus-infected skin’s inflammatory response. In addition, *M. furfur* can induce HaCaT cells to produce more lipid droplets, but POL reduces the lipid droplets induced by *M. furfur*. POL has no direct inhibitory effect on the growth of *M. furfur*, which benefits the balance of flora. Moreover, the micro-ecological environment and the immune system are too complicated to be illustrated, and we cannot cover them in one article. We hope to provide a breakthrough in the future.

## Figures and Tables

**Figure 1 molecules-27-07520-f001:**
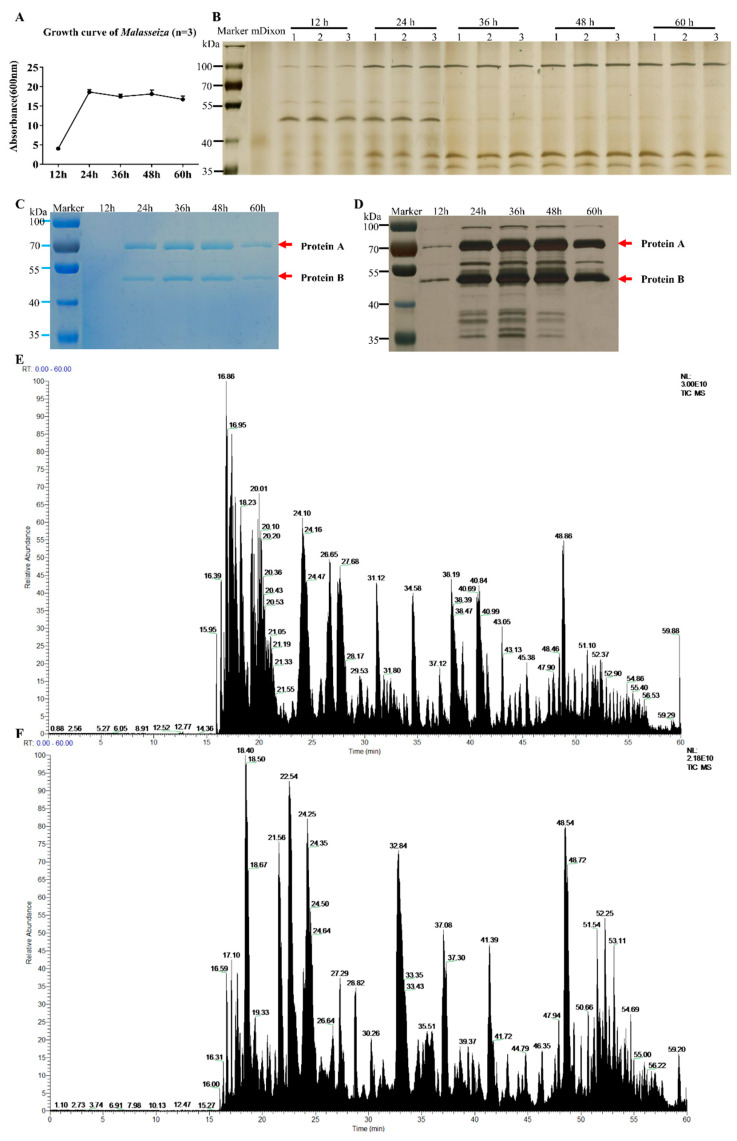
Growth curve and protein expression of *M. furfur*. (**A**) The growth curve of *M. furfur* was obtained from data of the absorbance (600 nm) of 12 h, 24 h, 36 h, 48 h, 60 h. (**B**–**D**) The Coomassie brilliant blue staining electropherogram of *M. furfur* cultivated in mDixon for 12 h, 24 h, 36 h, 48 h and 60 h. (**B**) Supernatant (n = 3), (**C**) cells (n = 1). (**D**) The silver staining of image (**C**). The two most abundant proteins are marked as Protein A and Protein B. (**E**) The total ion chromatogram of protein A. (**F**) The total ion chromatogram of protein B.

**Figure 2 molecules-27-07520-f002:**
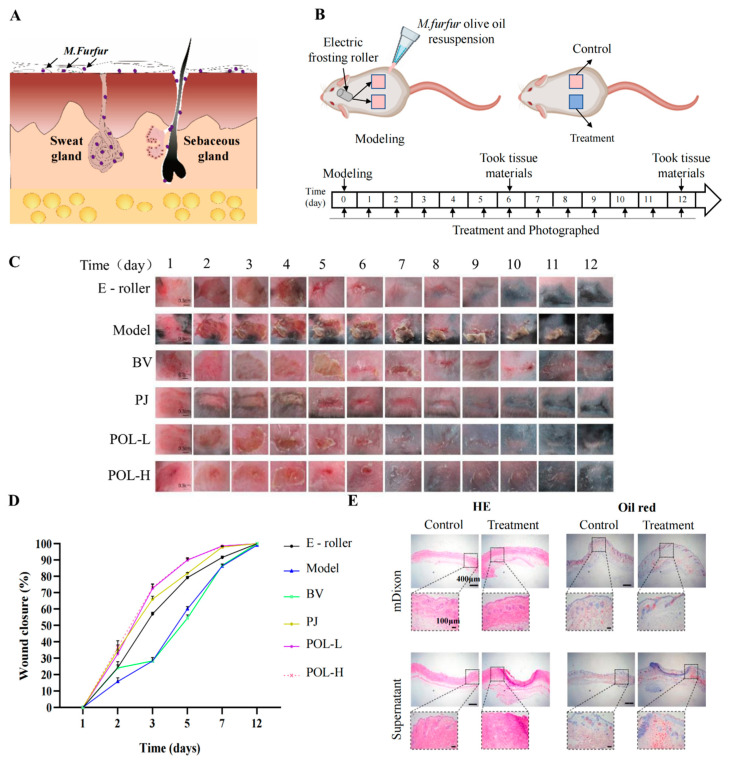
Polygalaxanthone III accelerates wound healing caused by *M. furfur* infection in mouse skin. (**A**) The distribution of *M. furfur* on the skin surface. (**B**) The experimental protocol of modeling and treatment. (**C**) Photos of the wound area about each group. Scale bar (b) = 0.3 cm. (**D**) The wound closure of each group at 2, 3, 5, 7, 12 days. (**E**) Images of HE and oil red O-stained tissue sections under the microscope. Scale bar = 400 μm (local enlarged view scale bar = 200 μm). E-roller: Electric frosting roller; Model: electric frosting roller and *M. furfur*; BV: Blank vehicle; PJ: Polygala japonica Houtt.; POL-L: low-dose POL; POL-H: high-dose POL.

**Figure 3 molecules-27-07520-f003:**
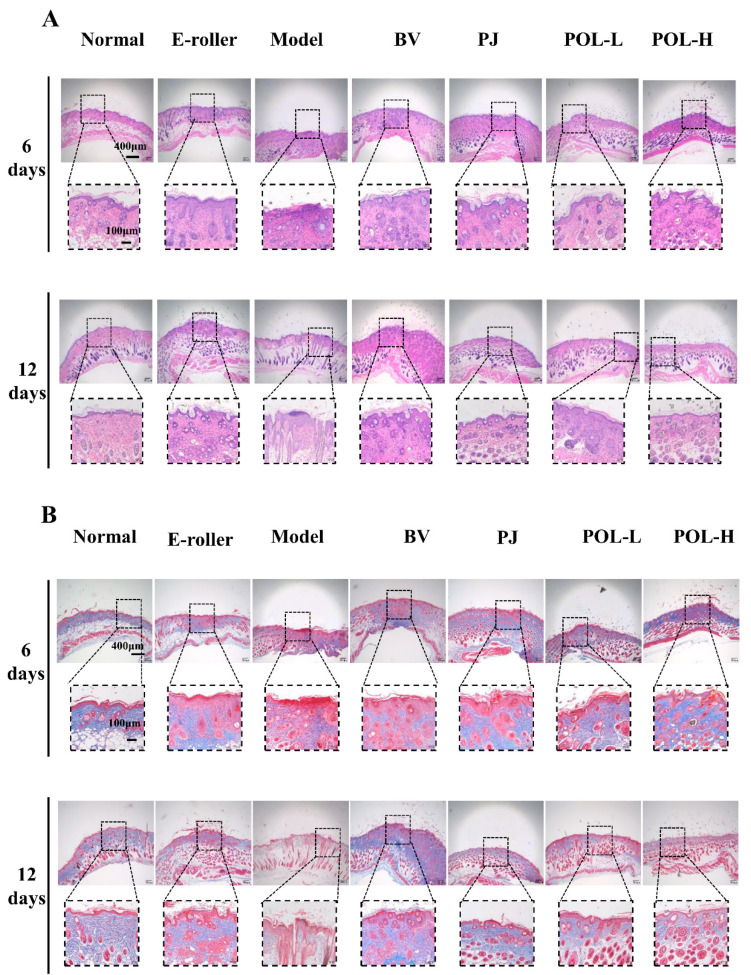
Images of HE and Masson staining of wound skin samples. (**A**) Pictures of HE-stained tissue sections under the microscope for each group at 6 days and 12 days (E-roller: Electric frosting roller; Model: electric frosting roller and *M. furfur*; BV: Blank vehicle; PJ: *Polygala japonica Houtt.*; POL-L: low-dose POL; POL-H: high-dose POL). Scale bar (**A**) = 400 μm (local enlarged view scale bar = 100 μm). (**B**) Images of Masson-stained tissue sections under the microscope for each group at 6 days and 12 days. Scale bar (**B**) = 400 μm (local enlarged view scale bar = 100 μm).

**Figure 4 molecules-27-07520-f004:**
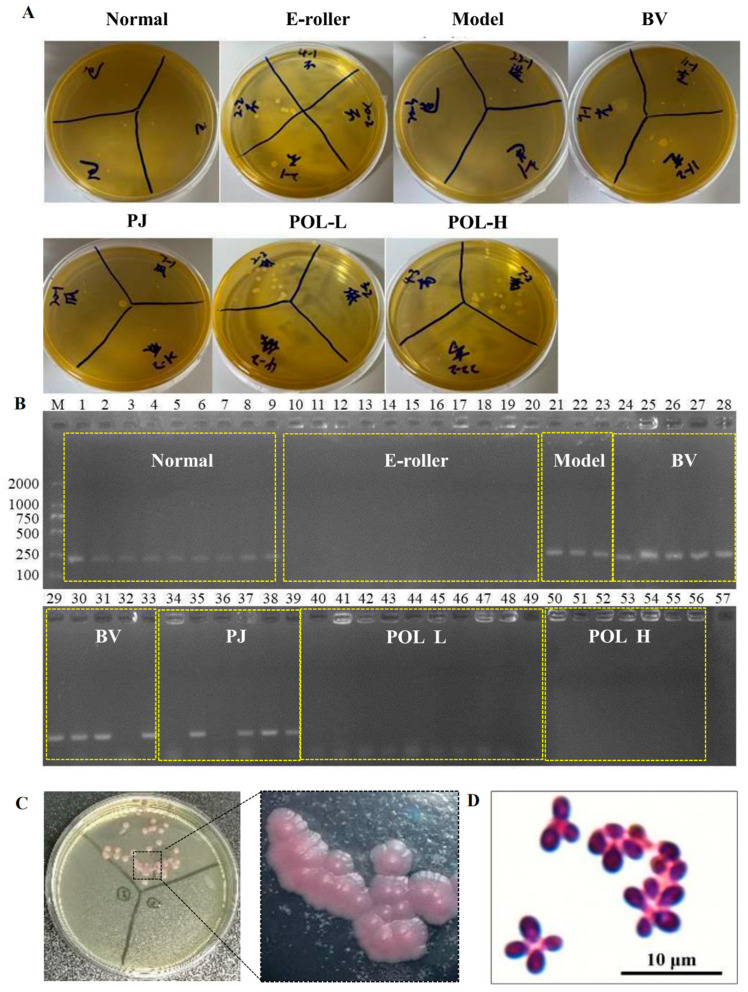
Identification and microscopic examination of *M. furfur* colonies in wound skin. (**A**) After 48 h of treatment, bacteria were taken from the wounds of the mice and smeared on the plate. (**B**) Identification of *M. furfur* monoclonal on plates smeared with wound bacteria by PCR (1–9 were from normal skin, 10–20 from the electric frosting roller group, 21–23 from the model group, 24–33 from the blank vehicle group, 34–39 from the *Polygala japonica Houtt* group (PJ), 40–49 from the low-dose POL group (POL L), 50–56 from the high-dose POL group (POL H), 57 as a negative control (sterilized water was used as a PCR template)) (E-roller: Electric frosting roller; Model: electric frosting roller and *M. furfur*; BV: Blank vehicle; PJ: *Polygala japonica Houtt.*; POL-L: low-dose POL, POL-H: high-dose POL). (**C**,**D**) Identification of *Malassezia* colonies by chromogenic medium (*Malassezia*-positive colonies will appear pink), and *Malassezia*-positive colonies examined under Nikon ECLIPSE NI-E.

**Figure 5 molecules-27-07520-f005:**
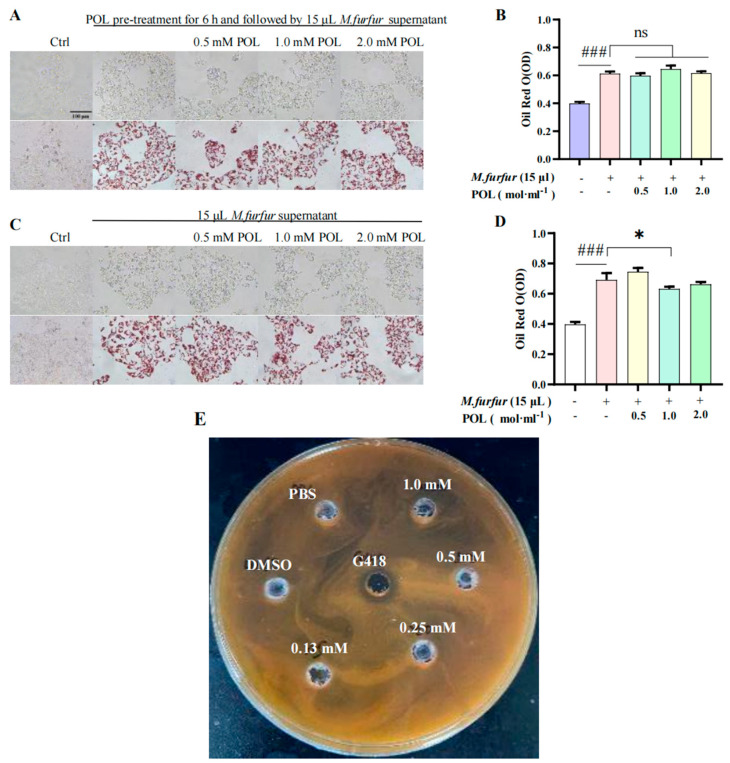
POL attenuated lipid droplet formation in *M. furfur*-stimulated HaCaT cells. Images of HaCaT cells (**A**) pre-treated with POL for 6 h, then followed by 15 μL *M. furfur* supernatant. (**B**) The number of lipid drops counted by oil red O in (**A**). (**C**) POL-treated *M. furfur*-stimulated HaCaT cells. (**D**) The number of lipid drops counted by oil red O in (**C**). (n = 3, ### *p* < 0.001 vs. Ctrl. ** p* < 0.05, ns means no significant differences, *p >* 0.05 vs. Model group). (**E**) The inhibition zone test of POL on the growth of *M. furfur*, G418 (Geneticin) as a positive inhibitor.

**Figure 6 molecules-27-07520-f006:**
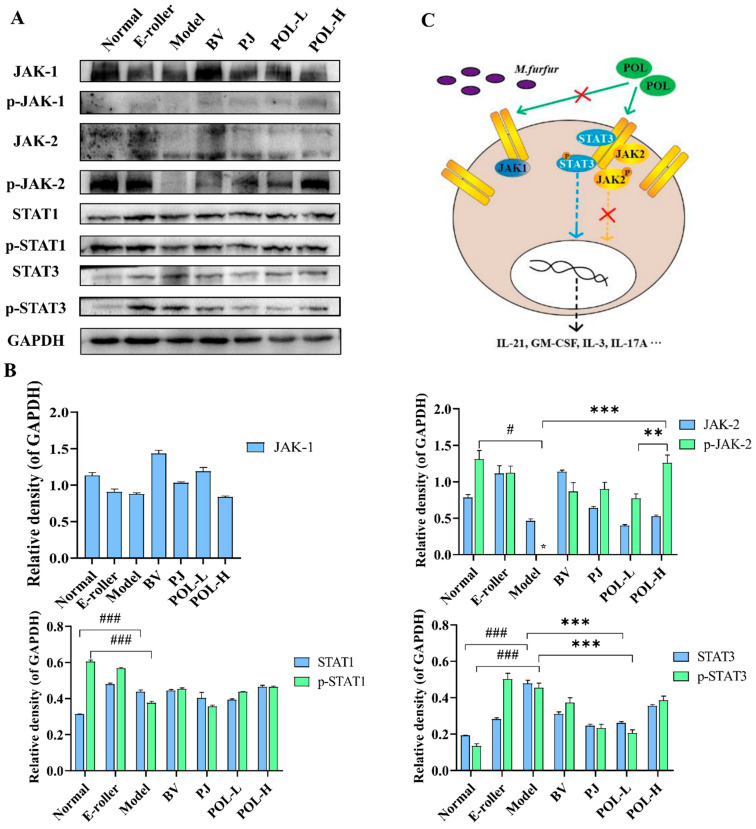
POL repaired the *M. furfur*-stimulated skin injury via STAT3 phosphorylated activation. (**A**) Western blot analysis of the expression of the JAK-STAT signaling pathway-related proteins and their quantitation in the *M. furfur*-stimulated mouse skin (E-roller: Electric frosting roller; Model: electric frosting roller and *M. furfur*; PJ: *Polygala japonica Houtt.*; POL-L: low-dose POL; POL-H: high-dose POL), GAPDH was used as the endogenous control, (**B**) the intensities of Western blots were measured by ImageJ software, n = 3, mean ± SD, # *p* < 0.05, ### *p* < 0.001, vs. Normal group. ** *p* < 0.01, *** *p* < 0.001, vs. Model group. (**C**) Potential signaling pathway of POL, POL would accelerate repair of the *M. furfur*-stimulated skin injury via STAT3 phosphorylated activation independent of JAK and associated with immune response.

## Data Availability

Not applicable.

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
