# Peer review of "Polygalaxanthone III, an Active Ingredient in Polygala japonica Houtt., Repaired Malassezia-Stimulated Skin Injury via STAT3 Phosphorylated Activation"

_molecules, 2022, doi:10.3390/molecules27217520_

Round 1

Reviewer 1 Report

The manuscript titled “Polygalaxanthone III, an anti-inflammatory ingredient in Polygala japonica Houtt. repaired the Malassezia - stimulated 3 Skin Injury via STAT3 phosphorylated activation” further deepens the knowledge on this natural product skin-healing activity and Malassezia spp. mechanism of action.

In order to improve the manuscript, the authors should address some points, namely:

1.    I would avoid the use of “an anti-inflammatory ingredient” in the title, as it suggests that the study is regarding inflammation;

2.    Please revise the English language and grammar (e.g. avoid the use of sentences such as “and so on”; “achieved wounds heal quickly”; “Weston blotting”; “accompanying with the human microbiome exploding”).

3.    Line 28, “Malassezia genus” could be used; Although M. furfur is the species used in the present study, it is not listed at the beginning of the introduction;

4.    Line 65-66, please revise, the sentence is hard to understand;

5.    Please revise the numbering of figures in the text (e.g. lines 87 and 90 do not correspond to figure 1 panel identification);

6.    Line 96, mass spectrometry analysis spectra data is not provided. The LC-MS should be described in the methodologies section;

7.    Line 287, ATCC® TIB-71 reference corresponds to RAW 264.7 mouse macrophages, and not to HaCaT cells, which are human keratinocytes;

8.    Figures 2 and 3 are identical, although presenting different captions. The same is observed in figures 4 and 5. For this reason, it was impossible to analyze the results reported by the authors, as well as the discussion and further conclusions;

Overall, the work presented by the authors is of high interest and has an interesting experimental design. The authors should provide all figures, so the results, discussion and conclusions can be provided.

Author Response

Manuscript Number: molecules-1966362  

Polygalaxanthone III, an anti-inflammatory ingredient in Polygala japonica Houtt. repaired the Malassezia - stimulated Skin Injury via STAT3 phosphorylated activation

Dear Editors and Reviewers,

Thank you for your letter and for the reviewers’ comments concerning our manuscript entitled “Polygalaxanthone III, an anti-inflammatory ingredient in Polygala japonica Houtt. repaired the Malassezia - stimulated Skin Injury via STAT3 phosphorylated activation” (ID: molecules-1966362). Those comments are all valuable and very helpful for revising and improving our paper, as well as the important guiding significance to our researches. We have studied comments carefully and have made correction which we hope meet with approval. The main corrections in the paper and the responds to the reviewer’s comments are on the next page.

We appreciate for Editors/Reviewers’ warm work earnestly, and hope that the correction will meet with approval.

Once again, thank you very much for your comments and suggestions.

Kind regards,

Xiaobin Yang

20 October 2022

Reviewer 2 Report

The article is interesting showing how the bioactive Polygalaxanthone and its potential anti-inflammation works against Malassezia However, its writing should be better improved.

1. The Introduction is quite too long due to many previous studies, it should be better short and more compact indicating the main problem and rationale. What are the worse side effects of chemical anti-fungal drugs? What will be the solution? How do the authors plan to solve the problem ?

2. What was the standard antifungal drug compared to and used as the positive control in the animal experiment?

These words and phrases highlighted with bright yellow have been grammatically re-written and correct.

3. There are several errors in English grammar (italics, tenses, ) that should be checked.

Examples: in vivo, in vitro, M. furfur,...etc.

2. Abstract

Malassezia is commensal and lipid-dependent yeasts in human skin, but they also have a patho-genic lifestyle associated with several common skin disorders, such as atopic dermatitis, eczema, and so on. Symptoms include red, itchy, and inflamed skin. We studied the growth characteristics and biochemical analyses of M.furfur, which showed that the protein contents were greater in 24h ex-tracts, then used them to infect C57BL/6 mice resulting in skin rupture. Polygalaxanthone III (POL), a more effective anti-inflammatory ingredient in Polygala japonica Houtt., was applied externally to the ulceration and achieved wounds heal quickly. POL could not inhibit the Malassezia action tested by the inhibition zone test, but affected the lipid droplet formation in HaCaT cells. The wound-healed molecular mechanisms may involve in STAT3 pathway according to the Weston Blotting results of skin tissues. Malassezia’s role in skin health is far from certain and there's no clean solution, so understanding Malassezia-associated skin disease development in general and sought for solutions is very important.

'

5. Conclusions

Eczema and seborrheic dermatitis (SD) et al are the results of hydrolysis of free fatty acids by and activation of the immune system through pattern recognition receptors, in inflammasomes, interleukin-1β and nuclear factor-κB [20]. The interaction of Malassezia, keratin-forming cells and the immune response against altered lipid composition in the skin play a crucial role in the pathogenesis of Malassezia dermatoses. Although the sequence of pathophysiological events regarding them is unclear, most sources consider the three main prerequisites to be: Malassezia colonization, sebaceous gland lipid secretion, and underlying immune system susceptibility [21]. The pathogenesis can be divided into five distinct phases [22]: sebaceous glands secrete lipids onto the skin surface; Malassezia colonizes areas covered with lipids; Malassezia secretes lipase, which produces free fatty acids and lipid peroxides that activate the inflammatory response; the immune system produces cytokines such as IL-1α, IL-1β, IL-2, IL-4, IL-6, IL-8, IL-10 IL-12, TNF-α, which stimulate keratinocyte proliferation and differentiation; and disruption of the skin barrier, leading to clinically evident erythema, pruritus and scaling.  This study successfully established a mouse skin model of infection and found that  POL can accelerate wound healing via STAT3 phosphorylated activation which plays an important role in fungal infected skin inflammatory response. In addition, M.furfur. can induce HaCaT cells to produce more lipid droplets, but POL reduces the lipid droplets induced by M.furfur. POL has no direct inhibitory effect on the growth of M.furfur., which benefits the balance of flora. Moreover, the micro-ecological environment and the immune system seem too complicated to be illustrated, and we cannot cover them in one article. Hope we can have a breakthrough in the future.

Author Response

(The authors gave the same response as above.)

Round 2

Reviewer 1 Report

In this version, the authors have successfully improved their manuscript based on the reviewers' comments. The detailed responses were clarifying, and the addition of correct images allowed the full analysis of the manuscript.

I would like to point out some details such as:

1. "The traditional Chinese medicine, Polygala japonica Houtt. (PJ), is a member of Polygala L. family.", in this sentence "Polygala" is the genus, the family is Polygalaceae.

2. CRL-2310 is the ATCC catalog number of CCD 1102 KERTr, which are BSL 2.

3. As a suggestion, doubling the size of figure 3 would allow the reader to better visualize the data. The images are small to fully see the morphological differences.

4. Was the toxicity of P. japonica previously tested in HaCaT? Should it be reported?

5.The results and discussion can still be improved, specially regaring the pathways involved in the results obtained in figure 6.

The manuscript should be revised for these critical details.

Author Response

Response to Reviewer 1 Comments

Point 1. "The traditional Chinese medicine, Polygala japonica Houtt. (PJ), is a member of Polygala L. family.", in this sentence "Polygala" is the genus, the family is Polygalaceae.

Response:

Thanks to the reviewer’s suggestions. The sentence of line 52-53 of the manuscript has been modified as “The traditional Chinese medicine, Polygala japonica Houtt. (PJ), is a member of Polygalaceae family”.

Point 2. CRL-2310 is the ATCC catalog number of CCD 1102 KERTr, which are BSL 2.

Response:

Thanks to the reviewer’s suggestions. You are right! We are ashamed for the mistake. The RRID of HaCaT cells is CVCL_0038 has been corrected in line 276.

Point 3. As a suggestion, doubling the size of figure 3 would allow the reader to better visualize the data. The images are small to fully see the morphological differences.

Response:

Thanks to the reviewer’s suggestions. The size of figure 3 has been doubled to see details.

Point 4. Was the toxicity of P. japonica previously tested in HaCaT? Should it be reported?

Response:

Thanks to the reviewer’s question. We are sorry for we didn’t test the toxicity of P. japonica in HaCaT. Then we explored literatures and wish to find some reports about it. But we found nothing. Since P. japonica is a TCM, which should be used after processed, such as boil, broil, poach, steam or others. Polygalaxanthone III (POL) as one of an active constituent of Polygala japonica, its content in Polygala japonica is low. If it is converted back to Polygala japonica, the dosage will too large. We think it is not particularly appropriate. By the way, we had tested the toxicity of POL in HaCaT cells, and found POL 31.25 ~1000 μM had no apparent toxicity to HaCaT at 24h and 48h.

Point 5.The results and discussion can still be improved, specially regaring the pathways involved in the results obtained in figure 6.

Response:

Thanks to the reviewer’s question. We have revised Results and Discussion according to reviewer's opinion, and the modifications made have been highlighted in the article. Please refer to the sections 2. Results and 3. Discussion.
